# Effect of Potato Tuber Exposure to UV-C Radiation and Semi-Product Soaking in Water on Acrylamide Content in French Fries Dry Matter

**Zygmunt Sobol** [1] **, Tomasz Jakubowski** [1,*] **and Magdalena Surma** [2]

1   Faculty of Production and Power Engineering, University of Agriculture, 30-149 Krakow, Poland; zygmunt.sobol@ur.krakow.pl
2   Faculty of Food Technology, University of Agriculture, 30-149 Krakow, Poland; magdalena.surma@ur.krakow.pl
*   Correspondence: tomasz.jakubowski@ur.krakow.pl

**Abstract:** This study aims to determine the effect of raw potato tubers' exposure to UV-C radiation and semi-products soaking in water on the content of acrylamide in the dry matter of French fries. The French fries were prepared from tubers of the Innovator variety of potato (*Solanum tuberosum* L.). Acrylamide contents were determined by HPLC-UV/Vis on a C-18 column after extraction of fried potatoes with acetonitrile. Potato tubers exposure to UV-C radiation caused an increase in acrylamide content and the soaking of semi-products in water caused a decrease in acrylamide content in the dry matter of French fries.

**Keywords:** French fries; UV-C; acrylamide; water soaking

---

## 1. Introduction

Due to the fact that acrylamide (AA) can be found in a wide assortment of food products consumed as part of an everyday diet, it can increase the risk of development of cancers in consumers from all age categories [1–5]. According to regulation no. 2017/2158 of 20 November 2017, in force since 11 April 2018, the European Commission set mitigation measures and benchmark levels for the reduction of the presence of acrylamide in food [4], whereas the U.S. Department of Health and Human Services Food and Drug Administration Center for Food Safety and Applied Nutrition [6] developed guidelines suggesting various approaches to reduce acrylamide levels. The mitigation measures recommended in the case of food products made of raw tubers of potato (*Solanum tuberosum* L.) include:

- choice of a suitable potato variety which, after meeting criteria set for a given type of food product, should have the lowest possible contents of acrylamide precursors, reducing sugars (fructose and glucose) and asparagine;
- choice of appropriate storage and transport conditions of potato tubers (i.e., temperature above 6 °C and relative air humidity inhibiting tuber ageing process).

Other recommended measures include suppression of tuber sprouting and monitoring levels of reducing sugars in the period of harvest and storage.

Considering French fries as well as other deep fried and oven-fried cut potato products, guidelines were provided as to the development of their recipe, production process, and necessary information for the end-users. The most important of these include: discarding immature tubers (separation depending on their density), removing slivers just after cutting, blanching semi-products (strips) depending on the quality traits of the raw material, preventing enzymatic browning, avoiding reducing sugars as

browning agents, keeping frying temperature at 160–175 °C, keeping oven baking temperature at 180–220 °C, heat treatment until golden-yellow color, avoiding over-baking/over-frying, and turning over the oven-baked products after 10 minutes or halfway through the total baking time. According to the aforementioned European Union (EU) regulation, the recommended mitigation measures are based on the current scientific and technical knowledge. Their efficacy is reflected in a reduced acrylamide content achieved without compromising the quality and safety of a food product. An overview of selected available literature providing results of investigations into the mitigation measures suggested for reducing acrylamide content in fried food products made of potato tubers is presented in Table 1.

**Table 1.** An overview of literature works providing results of investigations into the mitigation measures suggested for reducing acrylamide content in fried food products made of potato tubers.

| No. | Recommendations Represent Mitigation Measures* or not for Reducing Acrylamide Content in Fried Food Products Made of Potato Tubers | Literature Containing the Results of Research That Was the Source for Making Recommendations |
|---|---|---|
| 1. | Choice of suitable variety | Tajner-Czopek, Rytel, Nemś 2012 [7]; Belvoirpark Zürich, Kantonales Labor Zürich, Januar 2003 [8] |
| 2. | Choice of appropriate storage and transport conditions | Belvoirpark Zürich, Kantonales Labor Zürich, Januar 2003 [8]; Food Drink Europe 2011 [9] |
| 3. | Suppression of tuber sprouting | Food Drink Europe 2011 [9] |
| 4. | Monitoring levels of reducing sugars in the period of harvest and storage | Belvoirpark Zürich, Kantonales Labor Zürich, Januar 2003 [8]; Food Drink Europe 2011 [9] |
| 5. | Discarding immature tubers (separation depending on their density) | Food Drink Europe 2011 [9] |
| 6. | Removal of slivers just after cutting | Somsen, Capelle, Tramper 2004 [10]; Food Drink Europe 2011 [9] |
| 7. | Blanching and soaking in pure water and/or water with additives of semi-products (strips) depending on the quality traits of the raw material | Gertchen 2015 [11], Al-Asmar 2018 [12], Morales Capuano, Fogliano 2008 [13]; Mestdagh 2008 [14]; Bund für Lebensmittelrecht und Lebensmittelkunde e. V. AiF 209 ZBG. 2008 [15]; Belvoirpark Zürich, Kantonales Labor Zürich, Januar 2003 [8]; Kita 2006 [16] |
| 8. | Preventing enzymatic browning | Food Drink Europe 2011 [9] |
| 9. | Avoiding reducing sugars as browning agents | Higley i in. 2012 [17]; Food Drink Europe 2011 [9] |
| 10. | Keeping appropriate temperature of frying or/and baking | Tajner-Czopek i in. 2012 [18]; Sanny i in. 2013 [19]; Amrein 2006 [20]; Bund für Lebensmittelrecht und Lebensmittelkunde e. V. AiF 209 ZBG. 2008 [15]; Kita 2006 [16]; Belvoirpark Zürich, Kantonales Labor Zürich, Januar 2003 [8]; Food Drink Europe 2011 [9] |
| 11. | Heat treatment until golden-yellow color | Matthäus, Haase 2014 [21]; Food Drink Europe 2011 [9] |
| 12. | Avoiding over-frying/over-baking, turning over the oven-baked products every 10 minutes or halfway through the total baking time | Amrein 2006 [20]; Bund für Lebensmittelrecht und Lebensmittelkunde e. V. AiF 209 ZBG. 2008 [15]; Belvoirpark Zürich, Kantonales Labor Zürich, Januar 2003 [8]; Food Drink Europe 2011 [9] |
| 13. | Ratio of product surface area to volume | Gökmen V. Palazoglu T.K. 2009 [22]; Food Drink Europe 2011 [9] |

The food industry exploits physical methods based on electromagnetic waves to modify selected quality traits of raw material, semi-products, or finished products [23]. An effective method for the preservation of products made of biological raw materials is offered by ultraviolet radiation in the C band. An electromagnetic wave with the frequency of 253.7 nm is capable of reducing microbial population count, but at the same, by penetrating into an object, it can induce physicochemical changes in the activity level and availability of certain phytochemicals [12].

Pursuant to Appendix IV of the European Commission Regulation, the benchmark level of acrylamide in French fries and other deep fried or oven-fried ready-to-eat cut products made of potato tubers was set at 500 μg·kg$^{-1}$ per product. Considering that the risk posed to human health by acrylamide presence in food is high, that AA levels in food products is exceeded compared to the benchmark levels, and that the number and efficacy of mitigation measures are limited, investigations addressing these issues should be continued and novel methods meeting the criteria of mitigation measures should be implemented [24–26].

This study aims to determine the effect of raw potato tubers exposure to UV-C radiation and of semi-product soaking in water on the content of acrylamide in the dry matter of French fries.

## 2. Materials and Methods

Storage and laboratory experiments were conducted in the years 2016–2017 with potato tubers of Innovator variety. This potato variety is one of the most commonly used for French fries' production among both European and Polish enterprises. It is an early variety of B culinary type, with tubers having a regular shape (round-oval), shallow eyes, and mean starch content of 14.6%. Potatoes of the Innovator variety are resistant to flesh browning in the raw and cooked state and are suitable for storage. Tubers were cold-stored for 3 months in single layers on an openwork base. Storage temperature was kept at 10 °C, whereas relative air humidity was at 90%–95%. The experiment was performed in three repetitions (7-day intervals between the beginning of subsequent repetitions), and each repetition included three replications.

### 2.1. Potato Tuber Exposure to UV-C Radiation

The station used for biological material irradiation (Figure 1) with ultraviolet in the C band included a chamber equipped in a TUV UV-C NBV 15 W radiator [27,28]. Radiator construction enabled smooth regulation of the height above chamber bottom in the range from 0.4 to 1.0 m; the radiator was additionally equipped in a precise switch timer (model: AURATON 100). During irradiation, potato tubers were placed on a flat metal bottom with the size of 0.52 m$^2$. Various modes of UV-C irradiation were applied: (1, 3) irradiation on one side of the tuber for 30 min; (2, 4) irradiation on opposite sides of the tuber for 15 min each; and 0 mode, control sample (non-irradiated). The tubers were irradiated two days before semi-products were formed (1, 2) and before storage (3, 4). Ten potato tubers were irradiated in a single replication.

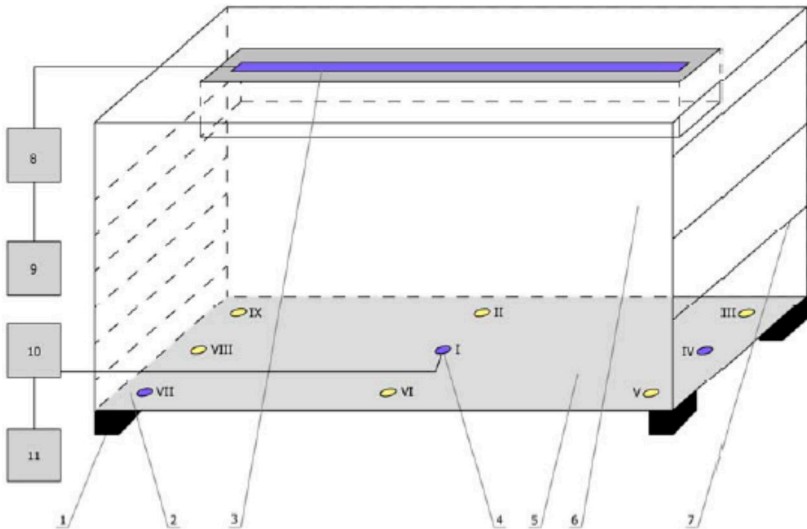

**Figure 1.** Test station 1 for UV-C irradiation of potato tubers: (1) supports, (2) bottom of the chamber, (3) UV-C radiator, (4) location of sensors (I-IX) of temperature, humidity, and UV-C intensity, (5) location of potato tubers, (6) chamber interior, (7) regulation of UV-C radiator height, (8) time-lag switch, (9) power supply, (10) multimeter, (11) data recorder.

### 2.2. Preparation of Samples for Heat Treatment

Semi-products for French fry production included potato strips (10 × 10 mm in cross-section and 60 mm in length). The number of strips used for frying was 14 in a single replication. The strips were cut out alongside the longitudinal tuber axis between the proximal and distal tuber end. Frying temperature was set at 170 °C and frying time at 15 min. The French fries were fried at once till they achieved sensory parameters appropriate for the finished product. The ratio of insert weight (semi-product) to frying oil weight was 1:15. The frying time was established in a separate preliminary experiment by analyzing sensory perceptions of evaluators. The excess of oil was removed from the

fried French fries in two stages: (1) on a grid, shaken base, and (2) on the absorptive paper material. To wash out reducing sugars from the semi-products, they were soaked in water with a temperature of 40 °C for 20 min (1). The non-soaked samples served as the control (0).

### 2.3. Preparation of Samples for Acrylamide Content Analysis

In order to precisely determine the effects of experimental factors (i.e., potato tubers exposure to UV-C radiation and semi-products soaking in water) on acrylamide content in the finished product, its content was determined with reference to French fries' dry matter (d.m.). To this end, the samples were dried in a laboratory drier at a temperature of 65 °C. The content of acrylamide (AA) was determined in the French fries tested. The samples were prepared according to Surma et al. [29] using a modified QuEChERS method. Briefly, 1 g of a homogenized sample of French fries was weighed into a 50 mL centrifuge tube. Then, 5 mL of water were added to the sample and a mixture of 10 mL of acetonitrile (MeCN) with 5 mL of hexane were added. The whole tube was vigorously shaken for 1 min, after which 1 g of NaCl and 4 g of $MgSO_4$ were added. This was followed by shaking for 1 min, and then the solution was finally centrifuged at 8700 RCF (relative centrifugal force) for 15 min. Afterwards, 6 mL of the supernatant were transferred to a 15 mL PP (polypropylene) tube containing 0.15 g primary and secondary amine (PSA), 0.3 g of C18, and 0.90 g of $MgSO_4$. After 1 min of shaking and 5 min of centrifugation at 5000 RCF, 4 mL of the supernatant were transferred into a 4 mL screw cup vial and evaporated to dryness under a stream of $N_2$. The residues were dissolved in 1 mL of MeCN and after filtration through a membrane filter were analyzed by HPLC-UV/Vis. Reagent blanks were prepared similarly, but without the sample. Each sample was prepared four times and analyzed three times.

### 2.4. Chromatographic Analysis Conditions

AA qualitative and quantitative analyses were performed using the HPLC-UV/Vis system (VWR HITACHI, LaChrom ELITE, Merck) according to Marconi et al. [30]. The chromatographic separation was performed at room temperature with C18 reversed-phase column (Lichrospher® 100, RP-18 end-capped, LiChroCART® 4 mm ID × 250 mm, 10 μm, Merck, Germany). AA was determined at 200 nm wavelength. All procedures were carried out isocratically using a mixture of 0.01 M sulfuric acid and water:methanol, 97.5:2.5. The flow rate was 0.7 mL $min^{-1}$.

The AA determined in this study was identified by the retention time. A series of standard solutions in acetonitrile were prepared within the range of 1–100 μg/mL. Each standard solution was prepared in triplicate. The limit of quantification (LOQ) was equal to 27.0 μg/kg. The results (AA concentration) were expressed as μg/kg d.m. (dry matter).

### 2.5. Chemicals

Acrylamide purum ≥98% was obtained from Sigma-Aldrich Chemie GmbH (Darmstadt, Germany). Magnesium sulphate anhydrous (pure for analysis) and sodium chloride p.a. were purchased from POCh SA (Gliwice, Poland). Acetonitrile, methanol, and hexane of HPLC grade for liquid chromatography LiChrosolve® and sulfuric acid were purchased from Merck KGaA (Darmstadt, Germany). PSA (primary and secondary amine) and C18 (octadecylsilane) SPE Bulk Sorbent were derived from Agilent Technologies (Santa Clara, CA, USA).

### 2.6. Statistical Analysis of Results

Results obtained were analyzed using Statistica 13.3 software at a significance level of $\alpha = 0.05$. Normality of distribution (Shapiro–Wilk test) and homogeneity of variance (Levene test) were analyzed. The analysis of variance was performed, and homogenous groups (marked in Tables 3 and 4 as ****) of variables were determined with the Duncan test. Histograms were plotted to depict results with the value of the standard deviation.

## 3. Results and Discussion

Results obtained from the Shapiro–Wilk test and the Levene test enabled the use of parametric statistical tests (Table 2). The analysis of variance conducted in double classification demonstrated that both of the experimental factors (i.e., UV-C irradiation of potato tubers (F = 90.2) and strips soaking in water (F = 720.4)) had a statistically significant effect on acrylamide content in French fries' dry matter. In addition, the interaction between the experimental factors proved to be statistically significant (F = 161.7).

**Table 2.** Results of analysis of variance for acrylamide content in Fresh fries' dry matter considering the main effects UV-C treatment and water soaking and their interaction.

| Factors | F-Value | *p*-Value |
|---|---|---|
| UV-C radiation {1} | 90.20 | 0.00 |
| Immersion of intermediates {2} | 720.40 | 0.00 |
| {1} × {2} | 161.75 | 0.00 |

Results of acrylamide content determination in French fries' dry matter demonstrate that it significantly (several times) exceeded the benchmark level stipulated in the EU Commission Regulation [4]. The analysis of acrylamide content in French fries' dry matter showed that potato tubers exposure to UV-C radiation contributed to an increase in its content compared to the control sample (Figure 2, Table 3). The distribution of homogenous groups (determined using Duncan's test) indicated that the effects caused by modes 4, 1, and 2 of potato tubers irradiation regarding acrylamide formation in French fries were the same (group 1) but differed significantly compared to the effects achieved with mode 3 of irradiation (group 3) and from effects observed in sample 0 (group 2) (Table 3).

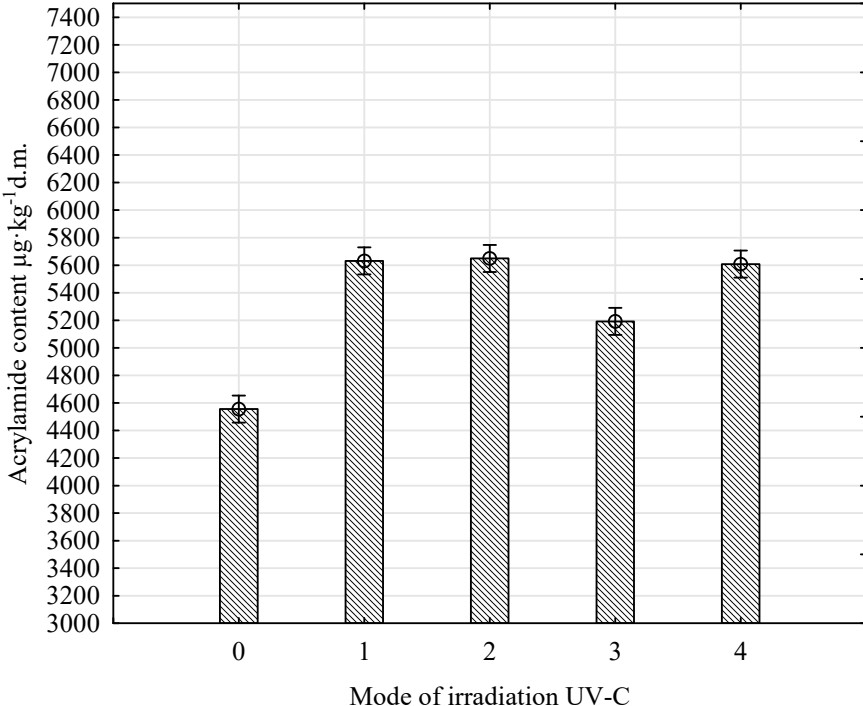

**Figure 2.** Effect of the mode of potato tubers irradiation on acrylamide content in French fries' dry matter (bars stand for standard deviation).

**Table 3.** Distribution of homogenous groups (Duncan's test). The effect of UV-C irradiation of potato tubers on acrylamide content in French fries' dry matter.

| UV-C Treatment | AA Content µg·kg$^{-1}$s.m. | Homogenous Groups | | |
| :---: | :---: | :---: | :---: | :---: |
| | | 1 | 2 | 3 |
| **0** | 4555.59 | | **** | |
| 3 | 5192.27 | | | **** |
| 4 | 5608.19 | **** | | |
| 1 | 5630.78 | **** | | |
| 2 | 5649.73 | **** | | |

The soaking of semi-products in water, aimed to reduce the contents of acrylamide formation precursors (elution of reducing sugars) in French fries, brought positive and expected outcomes. This treatment allowed reducing acrylamide content by ca. 1200 µg·kg$^{-1}$d.m. on average compared to the control samples (non-soaked) (Figure 3). Results obtained in the present study concerning reduction of acrylamide precursors content (reducing sugars in particular) are consistent with other findings (see [6,11–16]).

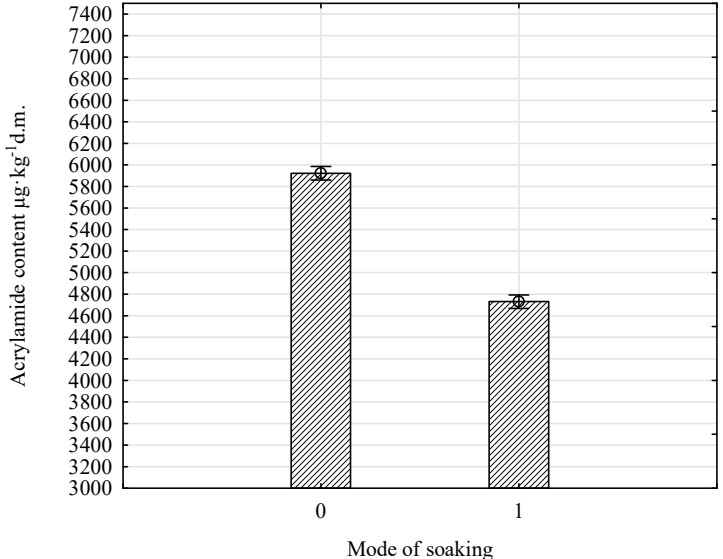

**Figure 3.** Effect of the mode of semi-products soaking in water on acrylamide content in French fries' dry matter (bars stand for standard deviation).

Potato tubers exposure to UV-C radiation increased acrylamide content in French fries' dry matter, mainly in the non-soaked samples. Distribution of homogenous groups in terms of the effect of UV-C irradiation of potato tubers and potato strips soaking in water on acrylamide content in French fries' dry matter indicated that the positive effect of reducing sugars elution was achieved mainly in the samples irradiated two days prior to processing (1, 2) and in the non-irradiated sample (0) (Table 4, Figure 4). The highest dynamics of acrylamide content decrease determined between the soaked samples and the control non-soaked sample was observed in the case of mode 2 of potato tubers irradiation. AA content decrease induced by this mode of irradiation reached 2831.01 µg·kg$^{-1}$d.m. (from 7065.24 µg·kg$^{-1}$d.m. for soaking mode 0 to 4234.23 µg·kg$^{-1}$d.m. for soaking mode 1) (Table 4). For irradiation mode 1, the decrease in AA content reached 1776.05 µg·kg$^{-1}$d.m. (from 6518.81 µg·kg$^{-1}$d.m. for soaking mode 0 to 4742.76 µg·kg$^{-1}$d.m. for soaking mode 1) (Table 4). A similar, significant difference in AA content decrease (1468.90 µg·kg$^{-1}$d.m.) was also demonstrated for mode 4 of potato tuber irradiation (from 6342.64 µg·kg$^{-1}$d.m. for soaking mode 0 to 4873.74 µg·kg$^{-1}$d.m. for soaking mode 1) (Table 4). Considering irradiation mode 3 and the variant with no irradiation, the differences in AA contents between the soaked and non-soaked samples were small compared to the effects induced by the other

irradiation modes tested. In the case of the non-irradiated samples, a small decrease was noted in AA content between the non-soaked and soaked samples, whereas in the case of irradiation mode 3 the tendency was opposite (i.e., AA content increased slightly) (Table 4).

**Table 4.** Distribution of homogenous groups (Duncan's test). The effect of UV-C irradiation of potato tubers and semi-products soaking in water on acrylamide content in French fries' dry matter.

| UV-C Irradiation | Semi-Products Soaking | Acrylamide Kontent $\mu g \cdot kg^{-1} s.m.$ | Homogenous Groups | | | | | | | |
|---|---|---|---|---|---|---|---|---|---|---|
| | | | 1 | 2 | 3 | 4 | 5 | 6 | 7 | 8 |
| 2 | 1 | 4234.23 | | | | | **** | | | |
| 0 | 1 | 4453.46 | | | | | | **** | | |
| 0 | 0 | 4657.74 | **** | | | | | | | |
| 1 | 1 | 4742.76 | **** | **** | | | | | | |
| 4 | 1 | 4873.74 | | **** | **** | | | | | |
| 3 | 0 | 5030.82 | | | **** | | | | | |
| 3 | 1 | 5353.72 | | | | | | | **** | |
| 4 | 0 | 6342.64 | | | | **** | | | | |
| 1 | 0 | 6518.81 | | | | **** | | | | |
| 2 | 0 | 7065.24 | | | | | | | | **** |

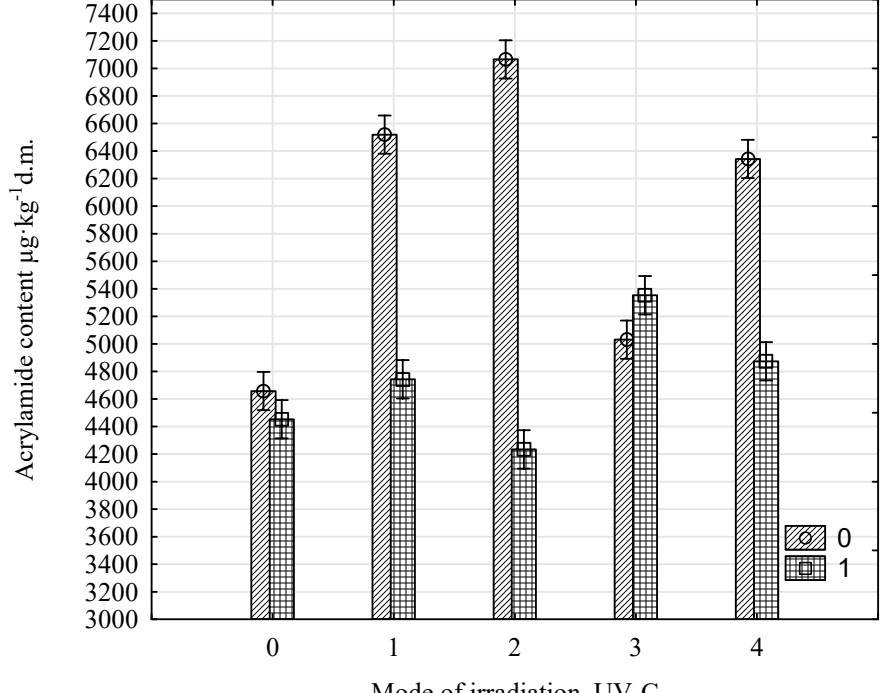

**Figure 4.** Effect of the interaction between the mode of potato tubers irradiation and the mode of semi-products soaking in water on acrylamide content in French fries' dry matter (bars stand for standard deviation).

Ultraviolet radiation with a wavelength of 253.7 nm pervades potato tubers only to a depth of ca. 2 mm; it interferes with the periderm and the adjacent flesh. The UV-C radiation induces a photochemical reaction in the form of photoisomerization, which can contribute to the transformation of flavonoids into their isomers due to the activity of photons [31–34]. Flavonoids are expected to protect a plant, but at the cellular level, to act as regulators of the cell cycle. Therefore, the UV-C radiation can influence the reduction of monosaccharides or initiate mechanisms aiding the process of sugars elution by immersion in water. The functioning of the described (putative) mechanisms could aid the process of reducing sugars elution from potato semi-products used to make French fries.

## 4. Conclusions

1.  Potato tubers' exposure to UV-C radiation caused an increase in acrylamide content in French fries' dry matter compared to the control non-irradiated sample.
2.  The soaking of semi-products in water caused a decrease in acrylamide content in French fries' dry matter.
3.  Potato tubers irradiation with UV-C caused an increase in acrylamide content in French fries' dry matter mainly when the semi-products were not soaked in water.
4.  The greatest decrease in acrylamide content was determined in French fries made of semi-products soaked in water. These semi-products were formed from potato tubers exposed to UV-C irradiation two days before processing.

## 5. Patents

Jakubowski T. Patent: The method and device for increasing the storage life of potato tubers with the participation of radiation UV-C (in polish; Sposób i urządzenie do zwiększania trwałości przechowalniczej bulw ziemniaczanych przy udziale promieniowania UV-C: P.419392, data zgłoszenia 07-11-2016).

Jakubowski T., Sobol Z. Patent: The method for modify the color of potato products and a device to modify the color of potato products (in polish; Sposób modyfikowania barwy wyrobów z ziemniaków i urządzenie do modyfikowania barwy wyrobów z ziemniaków: P.425887, data zgłoszenia 11-06-2018).

**Author Contributions:** Conceptualization, T.J. and Z.S.; methodology, T.J., Z.S., and M.S.; validation and formal analysis, T.J. and Z.S.; investigation, resources, and data curation, T.J., Z.S., and M.S.; writing—original draft preparation, writing—review and editing, and visualization, T.J. All authors have read and agreed to the published version of the manuscript.

**Funding:** This research received no external funding.

**Conflicts of Interest:** The authors declare no conflicts of interest.

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
