# Peer review of "Effect of Potato Tuber Exposure to UV-C Radiation and Semi-Product Soaking in Water on Acrylamide Content in French Fries Dry Matter"

_sustainability, doi:10.3390/su12083426_

Round 1

Reviewer 1 Report

The manuscript describes an interesting finding where UV irradiation can have a detrimental effect on potatoes. Overall, the manuscript looks good. However, I have the following comments that need to be addressed -

  1. References are not properly numbered. They should be properly numbered in the text, starting with #1.
  2. How long and at what intervals the UV-C exposure was done?
  3. Please describe all control, UV-irradiated, soaked, non-soaked groups in a separate table. It is hard and confusing to follow them in the text.  
  4. Page 6, lines 164-166, the authors mentioned -“The distribution of homogenous groups (determined using the Duncan’s test) indicated that the effects caused by modes 4, 1, and 2 of potato tubers irradiation regarding acrylamide formation in French fries were the same (group 1) but differed significantly compared to the effects achieved with mode 3 of irradiation (group 3) and from effects observed in sample 0 (group 2). According to the authors, “modes 1 and 3 received the same UV-C treatment”.  Why do they differ in acrylamide content? The authors should explain.

5. The authors should explain in more detail how the findings of this research can be applied to potatoes and French fries industry.

Author Response

Response to Reviewer 1 Comments

Thanks the Reviewer for comments

  1. References are not properly numbered. They should be properly numbered in the text, starting with #1.

Has been corrected

  1. How long and at what intervals the UV-C exposure was done?

Explanation

The experiment was divided into two periods - before storage and before processing. Prior to tuber storage, all samples were UV-C exposed for several days, and before processing the samples were UV-C irradiated two days before fries frying. This scope of the experiment was carried out over a time period of about 10 days.

  1. Please describe all control, UV-irradiated, soaked, non-soaked groups in a separate table. It is hard and confusing to follow them in the text.  

Explanation

The list of all factors of the experiment will be a repetition of the information contained in the chapter "Materials and methods".

  1. Page 6, lines 164-166, the authors mentioned -“The distribution of homogenous groups (determined using the Duncan’s test) indicated that the effects caused by modes 4, 1, and 2 of potato tubers irradiation regarding acrylamide formation in French fries were the same (group 1) but differed significantly compared to the effects achieved with mode 3 of irradiation (group 3) and from effects observed in sample 0 (group 2). According to the authors, “modes 1 and 3 received the same UV-C treatment”.  Why do they differ in acrylamide content? The authors should explain.

Explanation

Irradiation 1 and 3 was performed during 30 min on one side, but 1 concerns the procedure performed two days before tuber processing, and 3 before storage (i.e. 3 months before processing).

  1. The authors should explain in more detail how the findings of this research can be applied to potatoes and French fries industry.

Explanation

The experiment does not give unequivocal utilitarian value. Further research in this area should be focused on optimizing the exposure of potato tubers (maybe intermediates), to UV-C - in order to achieve the highest dynamics of leaching of reducing sugars (precursor of acrylamide formation) from intermediates, in French fries technology.

1) English language and style are fine/minor spell check required
English language and style was checked

Reviewer 2 Report

The manuscript entitle “Effect of potato tuber exposure to UV-C radiation and of semi-product soaking in water on acrylamide content in French fries dry matter” described acrylamide content in dry matter of French fries prepared from raw potato tubers exposed to UV-C radiation and of semi-product soaking in water. Experiment were designed well and results are interesting. I strongly believe the paper will be provide valuable information to both reader and researchers.

Author Response

Response to Reviewer 2 Comments

Thanks the Reviewer for comments

1) English language and style are fine/minor spell check required
English language and style was checked

Reviewer 3 Report

In the paper Authors present results on the effects of UV-C and water soaking pre-treatments on the acrylamide content of fried potatoes. This technological approach was the object of two patents specified at the end of the paper. The paper needs revisions listed in the following comments.

Abstract: L15: It would be preferable to be more precise. A suggestion: “Acrylamide contents were determined by HPLC-UV-Vis on a C-18 column after extraction of fried potatoes with acetonitrile.”

L16: Delete the phrase “Results obtained ……. level of α=0.05.” Instead, be more precise on the results obtained, referring also to the effect of water soaking and to the two types of UV-C application. Add also a conclusion phrase.

Keywords:  add “water soaking”

Introduction:

Table 1: In order to improve readability of the table, use in the column “Literature…” for each reference for more than 2 authors the style: First Author et al, YEAR [No] and acronyms for the Institutions, YEAR [No].

Materials and Methods section includes all the points indicated under section 3. Results. This section has to be re-organized. A suggestion: 1. Potatoes, 2. Chemicals (point 3.5), 3. Experimental plan (to be written), 4. UV-C treatment (point 3.1), 5. Water soaking treatment (L107-108), 6. Frying (point 3.2), 7. Acrylamide determination: 7.1 extraction (point 3.3), 7.2 HPLC analysis (point 3.4), 8. Statistical analysis (point 3.5). Hence at L145 the new title of the section is 3. Results and Discussion, and at L217 4. Conclusions.

Frying: please specify the type of oil used for frying and if it was renewed for each sample and replication

L113: please specify if you used the protocol of an official method; otherwise specify the time of drying and/or till constant weight.  In addition, it is not clear if a specific analysis for dry matter had been carried out and the extraction of acrylamide had been done on potatoes soon after frying or the whole fried potatoes had been dried prior to extraction.

Line 128 add reference number instead of Year for reference Marconi et al.

Chromatographic analysis: Add at the end of the section the criteria for acrylamide identification in the chromatogram, the concentration of standard solutions for calibration curve, the limit of quantitation and how the results are expressed.

Table 2: Title of the Table has to be rephrased. A suggestion: Results of analysis of variance for acrylamide content in Fresh fries dry matter considering the main effects UV-C treatment and water soaking and their interaction. As for table headings the suggestion is to use “Factors” instead of “Qualitv predictor and ineraction”, “F-value” instead of “Value of F-Snedecor test” and “P-value” instead of “Probabilyty test” Please delete the raw regarding “Free word” as  this is absolutely not to the point.

Table 3.  The code number of UV-C radiation could be replaced by a description; for example: for number 3 “one side, 30 min, before storage” and so on. This change will make clearer the reading of the table. As for columns’ heading “UV-C treatment” instead of “UV-C radiation”, “AA content – μg/kg d.m.” instead of “Acrylamide kontent μg kg-1s.m.”; please show data as mean ± standard error or standard deviation.

L164-167: please rewrite replacing the code of each UV-C treatment with its description, in order to make to be more straight to the point and to improve sentence understanding

Figure 2, 3 and 4: Y-axis: “d.m.” instead of “s.m.”; specify in the title of the Figure what bars stand for (standard error of the mean, cumulative standard error, deviation standard)

L176-178:  please write all the references in the format First Author et al, YEAR [No] and acronyms for the Institutions, YEAR [No]

Table 4.   similarly to what suggested for Table 3, the code number of UV-C treatment could be replaced by the description already used in Table 3, “UV-C treatment” instead of “UV-C irradiation”, “AA content – μg/kg d.m.” instead of “Acrylamide kontent  μg kg-1s.m.”; please show data as mean ± standard error or standard deviation.

Conclusions: delete the phrase “This section is not mandatory, but can be added to the manuscript if the discussion is unusually long or complex”. Instead, conclude giving a suggestion for producers of the best combination UV-C treatment and water soaking.

References in the text: According to the instruction “References must be numbered in order of appearance in the text (including table captions and figure legends) and listed individually at the end of the manuscript” and not listed in alphabetical order and then numbered. Please order the reference list in order of appearance in the manuscript, renumber each reference and make the suitable corrections in the text and table 1.

In addition:

L14, L29:  please write “Solanum tuberosum” in Italics, as it is in Latin

L91: “semi-products were prepared” instead of “semi-products were formed”

L212: please delete “,i.a.,”

L246: insert a space between “Fries” and “Coatings”. write “Coatings” in italics, as it is the Journal name

L271, L287, L324: please change capitals to lower case

References number 2, 5, 9, 10, 11, 13, 14, 17, 18, 19,20, 22, 28, 29,30, 31, add doi number at the end

References numbers 12 ,24,26: move doi number at the end

Reference 19: there are two journals cited for the same paper. Please check and correct.

References number 5, 7, 10, 11, 12, 13, 14, 17, 18, 19, 22, 23,26,27,28,29,31,32 use the Abbreviated Journal Name

Author Response

Response to Reviewer 3 Comments

Thanks the Reviewer for comments

  • Abstract: L15: It would be preferable to be more precise. A suggestion: “Acrylamide contents were determined by HPLC-UV-Vis on a C-18 column after extraction of fried potatoes with acetonitrile.”

We agree - changed in the content of the manuscript

  • L16: Delete the phrase “Results obtained ……. level of α=0.05.” Instead, be more precise on the results obtained, referring also to the effect of water soaking and to the two types of UV-C application. Add also a conclusion phrase.

We agree - changed in the content of the manuscript

  • Keywords:  add “water soaking”

We agree - changed in the content of the manuscript

  • Introduction: Table 1: In order to improve readability of the table, use in the column “Literature…” for each reference for more than 2 authors the style: First Author et al, YEAR [No] and acronyms for the Institutions, YEAR [No].

Explanation

Other Reviewers did not raise this issue. The designation adopted refers to the method of citing the literature used throughout the work (this should be treated as a justified repetition). The presented data result directly from the cited references. Providing acronyms for institutions is difficult because scientists represent different universities and institutes. Entering the year of publication is important because it indicates the period of publication (data timeliness).

  • Materials and Methods section includes all the points indicated under section 3. Results. This section has to be re-organized. A suggestion: 1. Potatoes, 2. Chemicals (point 3.5), 3. Experimental plan (to be written), 4. UV-C treatment (point 3.1), 5. Water soaking treatment (L107-108), 6. Frying (point 3.2), 7. Acrylamide determination: 7.1 extraction (point 3.3), 7.2 HPLC analysis (point 3.4), 8. Statistical analysis (point 3.5). Hence at L145 the new title of the section is 3. Results and Discussion, and at L217 4. Conclusions.

Explanation

We agree - changed in the content of the manuscript

Chapter 3 (Results and discussion) was introduced, numbering of chapter 2 was redrafted.

The plan of experiments results directly from the order of sections in this chapter. Creating a separate chapter "experimental plan" would repeat the content of the methodology.

  • Frying: please specify the type of oil used for frying and if it was renewed for each sample and replication.

Explanation:

Rapeseed oil was used as the fryer, which was changed (renewed) for each sample and replication

  • L113: please specify if you used the protocol of an official method; otherwise specify the time of drying and/or till constant weight.  In addition, it is not clear if a specific analysis for dry matter had been carried out and the extraction of acrylamide had been done on potatoes soon after frying or the whole fried potatoes had been dried prior to extraction.

Explanation

Drying of the samples was carried out until a constant sample weight was obtained. The French fries drying procedure was performed for all samples before AA extraction.

  • Line 128 add reference number instead of Year for reference Marconi et al.

We agree - changed in the content of the manuscript

  • Chromatographic analysis: Add at the end of the section the criteria for acrylamide identification in the chromatogram, the concentration of standard solutions for calibration curve, the limit of quantitation and how the results are expressed.

We agree - changed in the content of the manuscript

  • Table 2: Title of the Table has to be rephrased. A suggestion: Results of analysis of variance for acrylamide content in Fresh fries dry matter considering the main effects UV-C treatment and water soaking and their interaction. As for table headings the suggestion is to use “Factors” instead of “Qualitv predictor and ineraction”, “F-value” instead of “Value of F-Snedecor test” and “P-value” instead of “Probabilyty test” Please delete the raw regarding “Free word” as  this is absolutely not to the point.

We agree - changed in the content of the manuscript

  • Table 3.  The code number of UV-C radiation could be replaced by a description; for example: for number 3 “one side, 30 min, before storage” and so on. This change will make clearer the reading of the table. As for columns’ heading “UV-C treatment” instead of “UV-C radiation”, “AA content – μg/kg d.m.” instead of “Acrylamide kontent μg kg-1s.m.”; please show data as mean ± standard error or standard deviation.

We agree - changed in the content of the manuscript*

*Explanation

The numbering in Table 3 refers to the specific designation of the experiment combination. Entering complete descriptions for the combinations would be a repetition. Table 3 shows the average values and the value of standard deviation is shown in Figure 2. The introduction of the standard error value in Table 3 would be a repetition.

  • L164-167: please rewrite replacing the code of each UV-C treatment with its description, in order to make to be more straight to the point and to improve sentence under standing

Explanation

The designations of the individual combinations of the experiment are presented in the methodology. We tried to avoid repetition.

  • Figure 2, 3 and 4: Y-axis: “d.m.” instead of “s.m.”; specify in the title of the Figure what bars stand for (standard error of the mean, cumulative standard error, deviation standard).

We agree - changed in the content of the manuscript

(bars stand for standard deviation)

  • L176-178:  please write all the references in the format First Author et al, YEAR [No] and acronyms for the Institutions, YEAR [No]

We agree - changed in the content of the manuscript*

*Providing acronyms for institutions is difficult because scientists represent different universities and institutes.

  • Table 4.   similarly to what suggested for Table 3, the code number of UV-C treatment could be replaced by the description already used in Table 3, “UV-C treatment” instead of “UV-C irradiation”, “AA content – μg/kg d.m.” instead of “Acrylamide kontent  μg kg-1s.m.”; please show data as mean ± standard error or standard deviation.

We agree - changed in the content of the manuscript*

*Explanation

The numbering in Table 4 refers to the specific designation of the experiment combination. Entering complete descriptions for the combinations would be a repetition. Table 4 shows the average values and the value of standard deviation is shown in Figure 5. The introduction of the standard error value in Table 4 would be a repetition.

  • Conclusions: delete the phrase “This section is not mandatory, but can be added to the manuscript if the discussion is unusually long or complex”. Instead, conclude giving a suggestion for producers of the best combination UV-C treatment and water soaking

We agree - changed in the content of the manuscript*

*Explanation

The experiment does not give unequivocal utilitarian value. Further research in this area should be focused on optimizing the exposure of potato tubers (maybe intermediates), to UV-C - in order to achieve the highest dynamics of leaching of reducing sugars (precursor of acrylamide formation) from intermediates, in French fries technology.

  • References in the text: According to the instruction “References must be numbered in order of appearance in the text (including table captions and figure legends) and listed individually at the end of the manuscript” and not listed in alphabetical order and then numbered. Please order the reference list in order of appearance in the manuscript, renumber each reference and make the suitable corrections in the text and table 1.

We agree - changed in the content of the manuscript

  • In addition:

L14, L29:  please write “Solanum tuberosum” in Italics, as it is in Latin

We agree - changed in the content of the manuscript

L91: “semi-products were prepared” instead of “semi-products were formed”

We agree - changed in the content of the manuscript

L212: please delete “,i.a.,”

We agree - changed in the content of the manuscript

L246: insert a space between “Fries” and “Coatings”. write “Coatings” in italics, as it is the Journal name

We agree - changed in the content of the manuscript

L271, L287, L324: please change capitals to lower case

We agree - changed in the content of the manuscript

References number 2, 5, 9, 10, 11, 13, 14, 17, 18, 19,20, 22, 28, 29,30, 31, add doi number at the end

We agree - changed in the content of the manuscript

References numbers 12 ,24,26: move doi number at the end

We agree - changed in the content of the manuscript

Reference 19: there are two journals cited for the same paper. Please check and correct.

We agree - changed in the content of the manuscript

References number 5, 7, 10, 11, 12, 13, 14, 17, 18, 19, 22, 23,26,27,28,29,31,32 use the Abbreviated Journal Name

 We agree

1) English language and style are fine/minor spell check required
English language and style was checked